# From conservation to structure, studies of magnetosome associated cation diffusion facilitators (CDF) proteins in Proteobacteria

Noa Keren-Khadmy[1,2,3], Natalie Zeytuni[1,2,3], Nitzan Kutnowski[1], Guy Perrière[4], Caroline Monteil[4,5], Raz Zarivach[1,2,3]*

**1** Department of Life Sciences, Ben-Gurion University of the Negev, Beer-Sheva, Israel, **2** The National Institute for Biotechnology in the Negev, Ben-Gurion University of the Negev, Beer-Sheva, Israel, **3** Ilse Katz Institute for Nanoscale Science & Technology, Ben-Gurion University of the Negev, Beer-Sheva, Israel, **4** Laboratoire de Biométrie et Biologie Evolutive, UMR CNRS 5558, Université de Lyon, Villeurbanne Cedex, France, **5** CNRS, CEA, Aix-Marseille Université, UMR7265 Biosciences and Biotechnologies Institute of Aix-Marseille, Saint Paul lez Durance, France

* zarivach@bgu.ac.il

## Abstract

Magnetotactic bacteria (MTB) are prokaryotes that sense the geomagnetic field lines to geo-locate and navigate in aquatic sediments. They are polyphyletically distributed in several bacterial divisions but are mainly represented in the Proteobacteria. In this phylum, magneto-tactic Deltaproteobacteria represent the most ancestral class of MTB. Like all MTB, they synthesize membrane-enclosed magnetic nanoparticles, called magnetosomes, for mag-netic sensing. Magnetosome biogenesis is a complex process involving a specific set of genes that are conserved across MTB. Two of the most conserved genes are *mamB* and *mamM*, that encode for the magnetosome-associated proteins and are homologous to the cation diffusion facilitator (CDF) protein family. In magnetotactic Alphaproteobacteria MTB species, MamB and MamM proteins have been well characterized and play a central role in iron-transport required for biomineralization. However, their structural conservation and their role in more ancestral groups of MTB like the Deltaproteobacteria have not been estab-lished. Here we studied magnetite cluster MamB and MamM cytosolic C-terminal domain (CTD) structures from a phylogenetically distant magnetotactic Deltaproteobacteria species represented by BW-1 strain, which has the unique ability to biomineralize magnetite and greigite. We characterized them in solution, analyzed their crystal structures and compared them to those characterized in Alphaproteobacteria MTB species. We showed that despite the high phylogenetic distance, MamB$_{BW-1}$ and MamM$_{BW-1}$ CTDs share high structural simi-larity with known CDF-CTDs and will probably share a common function with the Alphapro-teobacteria MamB and MamM.

## Introduction

Magnetotactic bacteria (MTB) are unique gram-negative aquatic prokaryotes, that biosynthe-size magnetic iron mineral nanoparticles composed of magnetite ($Fe_3O_4$) or greigite ($Fe_3S_4$),

**Data Availability Statement:** All pdb files are available from the protein data bank database (accession number(s) 6QFJ, 6QEK).

**Funding:** The authors of this work are supported by the Israel Ministry of Science, Technology, and Space (R.Z.), the Israel Science Foundation (grant No. 167/16; R.Z.) and the European Molecular Biology Organization and CMST COST Action CM1306 (R.Z.). The funders had no role in study design, data collection and analysis, decision to publish, or preparation of the manuscript.

**Competing interests:** The authors have declared that no competing interests exist.

**Abbreviations:** MTB, magnetotactic bacteria; CDF, cation diffusion facilitator; CTD, c-terminal domain; Mam, magnetosome-associated membrane; TMD, trans-membrane domain; BW-1, *Desulfamplus magnetovallimortis* strain BW-1 SEC, size-exclusion chromatography; MALS, multi-angle light scattering; SAXS, small-angle X-ray scattering; PDB, protein data bank; RMSD, Root-mean-square deviation; C.B, central-binding site; P.B, peripheral metal-binding-site; PEG, polyethylene glycol; LB, Luria broth; RP, bacterial riboproteins; HMM, hidden Markov mode; PMSF, phenylmethylsulfonyl fluoride; ESRF, European Synchrotron Radiation Facility.

in a special membranous organelle called a magnetosome [1]. Chains of aligned magnetosomes [2–4] give a permanent magnetic moment to the cell and assist the bacteria to reach their suitable oxic-anoxic transition zone habitat [5,6]. Most of the genes crucial for the magnetosome biogenesis are clustered in a conserved region within their genomes [2,7–11].

Among these genes, two highly-conserved cation diffusion facilitator (CDF) protein homologs exist; *mamB* and *mamM* [12,13]. These genes are conserved in the Proteobacteria, Nitrospira and *Ca.* Omnitrophica phyla, in which magnetosomes are composed mostly from magnetite. In Proteobacteria, some MTB from the genetically distant Deltaproteobacteria class, *Desulfamplus magnetovallimortis* strain BW-1, can biomineralize both greigite and magnetite according to the environmental conditions [1]. It was suggested that such ability associates with additional dedicated clusters of genes, among them *mamB* and *mamM*, which may be directly related to greigite synthesis.

CDFs are utilized by MTB to transport iron cations from the bacterial cytoplasm into the magnetosome lumen during nanoparticle biomineralization [3,4,14]. Like most CDFs, the magnetosome membrane-associated (Mam) proteins MamB and MamM share a common two-domain architecture consisting of a trans-membrane domain (TMD) and a cytosolic C-terminal domain (CTD) [15,16]. The structures of FieF (YiiP), an iron/zinc transporter from *Escherichia coli* (*E. coli*), and *Shewanella oneidensis* MR-1, were determined in a zinc-bound state and assembled as a homodimer. Each monomer within the TMD is composed of six alpha-helices [15,17], while the CTD composed of two alpha-helices, and three beta-strands [17–19]. There are a few CDF activation transport mechanisms that were suggested. One suggestion is that the metal-ion binding to the CTD induces its dimerization, while a second model suggests that the metal-ion binding to the CTD induces a tighter-CTD packing [18,20,21]. Overall, the two models share the same suggested CTD conformational change, followed by TMD conformational change that activates iron transport through the TMD. Moreover, the current working model suggests that the CTD has a regulatory role [14,18,21] and that the TMD promotes metal ion transport by exploiting chemiosmotic gradients [14,17,22,23] via TMD alternation between inward- to outward-facing [24]. The vast majority of CDF structural studies are concentrated at the soluble CTD. CDF-CTD structures from *Thermotoga maritima* (TM0876$_{206-306}$) and MamM from the magnetotactic Alphaproteobacteria *Magnetospirillum gryphiswaldense* MSR-1 (MamM$_{MSR-1}$) were determined in their apo-form [21,25]. The CDF-CTD structures of CzrB from *Thermus thermophilus* and MamB from the *Magnetospira* sp. QH-2 (MamB$_{QH-2}$) were determined at the apo and ion-bound forms [14,26], whereas the full CDF structures of YiiP from *E. coli* and *Shewanella oneidensis* MR-1 were determined only in the ion-bound form [15,24,26,27]. Overall, all CTD structures present a stable V-shape dimer and share a similar metallochaperone-like fold [14,17,18,21,25,28]. All CDF-CTD dimers have a single dimerization interface located at the bottom of the V-shaped fold [14,17,21,24–26]. Such a small dimerization interface (193–400 Å$^2$) [14,18,24,29] mainly rests on a hydrophobic interaction, substitution-mutations within these residues leads to alternated dimeric packing that impaired protein function [12,20].

Previous studies on MTB focused on the characterization of MamB and MamM in Alphaproteobacteria (*Magnetospirillum sp.* strains MSR-1 and AMB-1, and *Magnetospira sp.* QH-2) and well demonstrated their active role in iron-transport [12,29,30]. Furthermore, it was shown that MamB has an additional key role in magnetosome vesicle invagination and magnetite nucleation that cannot be compensated by MamM activity [12,29,30]. Such roles were not investigated in distantly MTB species with peculiar biomineralization like that of some dual magnetite-greigite producing Deltaproteobacteria species.

Here we studied magnetite cluster MamB and MamM CTD structures from a phylogenetically distant magnetotactic Deltaproteobacteria species, BW-1 (MamB$_{BW-1}$ and MamM$_{BW-1}$).

We characterized $MamB_{BW-1}$ and $MamM_{BW-1}$ CTDs in solution, analyzed their crystal structures and compared them to CDFs in magnetotactic Alphaproteobacteria. This is the first structural study of MamB and MamM from the same bacteria strain, allowing a direct comparison of two CDFs in the same organelle. We showed that despite the high evolutionary divergence and the phenotypic differences between magnetotactic Alphaproteobacteria and Deltaproteobacteria, $MamB_{BW-1}$ and $MamM_{BW-1}$ CTDs share high structural similarity with known CDF-CTDs. This supports the current hypothesis that they share a common function with Alphaproteobacteria MamB and MamM. Our results also indicate that MamB could have emerged from a duplication of MamM, which means that the MamB ancestor was likely an iron transporter and that its role in magnetosome vesicle invagination and magnetite nucleation emerged secondarily in a magnetotactic ancestral Deltaproteobacteria.

## Results and discussion

### MamB and MamM CTDs from *Desulfamplus magnetovallimortis* strain BW-1 expression and characterization

To obtain structural and biochemical information, recombinant His-tagged $MamB_{BW-1}$ and $MamM_{BW-1}$ CTDs (residues 190–270 and 213–293 respectively) were over- produced in *E. coli* cells. Proteins were purified using affinity column followed by size-exclusion chromatography and found to be soluble as two oligomeric states, a monomer, and a dimer. To better characterize protein populations, size-exclusion chromatography with multi-angle light scattering (SEC-MALS) was used. According to the SEC-MALS, the correlated protein molecular weight ($M_W$) was calculated, 9.6 and 21 kDa for $MamB_{BW-1}$ and 11 and 25 kDa for $MamM_{BW-1}$ (S1B Fig).

To validate these results and to examine whether there is a correlation between protein concentration and oligomerization states, we continued with a small-angle X-ray scattering (SAXS) experiment. SAXS results showed that $MamB_{BW-1}$ scattering curves are similar at concentrations of 1 and 5 mg mL$^{-1}$, indicating that the molecular dimensions are independent of protein concentration over this concentration range. The radius of gyration (Rg) was ~ 1.8 nm, as obtained by the Guinier equation for dilute solutions. In contrast, $MamM_{BW-1}$ scattering curves differ at the same concentrations, showing Rg of ~ 1.87 nm and ~2.3 nm respectively. Such scattering differences can indicate that the molecular dimensions are dependent on protein concentration when at higher protein concentration dimer is preferred (S1 Table). These results differ from previous CDF studies in which stable dimers were observed at a concentration range of 1–28 mg mL$^{-1}$ [14,21,28]. Yet several hypotheses can support such findings. First, the suggested mechanism in which CTD metal-ion binding promotes CTD dimer assembly [31]. Despite that, we assume that metal-ions may support such interaction, but probably more factors are involved since other studies have shown that CTD assembles as a stable dimer even in the absence of metal ions [14,21,26,29]. Second, it was previously suggested that the full CDF dimer stabilization is dependent on the association between CTD and TMD. Therefore, in the absence of TMD, the stabilization of CTD-dimers might be lower [15,17]. Third, we assume that *in-vivo* MamB and MamM local-concentration might be higher as the ratio of protein to surface is higher than other membranes [32]. Such assumption is based on previous studies suggesting that MamB and MamM are initial landmark proteins to prime protein-complex formation as the first key step for magnetosome vesicle invagination [33,34]. Moreover, MSR-1 alphaproteobacteria MamB-MamM CTDs have previously shown to interact *in-vitro* and were suggested to assemble as heterodimers [30]. It was further proposed that the magnetosome lumen formed only after reaching critical size and composition of the multi-lipid-protein assembly that also involved MamQ and MamL [33]. In accordance

with this research and together with our SAXS results, we expect that in native conditions, $MamB_{BW-1}$ and $MamM_{BW-1}$ preferred oligomerization state is a hetro or homodimer.

## MamB and MamM crystal structural analysis

Next, we examined MamB and MamM CTD structural conservation and differences compared to homologous protein structures (Fig 1). For that, purified $MamB_{BW-1}$ and $MamM_{BW-1}$ CTDs were crystallized in different conditions (S2 Table). $MamB_{BW-1}$ with 6xHis-Tag crystallized with P1 space group and with six monomers per asymmetric unit (PDB code 6QFJ) and $MamM_{BW-1}$ crystallized with $P6_1$ space group and with two monomers per asymmetric unit (PDB code 6QEK) (S3 Table). Each $MamB_{BW-1}$ and $MamM_{BW-1}$ monomer consists of two alpha-helices and three beta-strands and is folded like-metallochaperone (Fig 2), similar to previous CDF-CTD crystal structures [14,18,21,24]. We calculated the Root-mean-square deviation (RMSD) between MamB and MamM CTDs from remoted phyla to probe the structural differences (S4 Table). A noticeably high structural similarity between all MamB and MamM structures was observed (Fig 2). $MamM_{BW-1}$ and $MamM_{MSR-1}$ share the highest structural similarities with RMSD of 0.66Å over 77 common backbone atoms and 1.12 Å over 127 backbone atoms for a monomer and dimer respectively; such values are related to an error value between similar crystals (Fig 1C).

## Dimerization and functional conservation

To examine the protein oligomeric states and the dimerization interfaces within the crystal structures, we studied the dimer V-shaped structure (Fig 3A). $MamM_{BW-1}$ dimerization interface relies mainly on the interactions between Pro278-Val283, Ser280-Ile282, Ile282-Ser280, and Val283-Pro278 and has a dimerization surface of 589 $Å^2$. The contacts include hydrophobic interactions and hydrogen bonds similar to the interactions that were observed earlier for $MamB_{QH-2}$, $MamM_{MSR-1}$ and other CTDs [14,18,21]. Such interface is higher than previous dimerization surfaces, Alphaproteobacteria $MamB_{QH-2}$ (497 $Å^2$) or $MamM_{MSR-1}$ (379 $Å^2$) and CzrB apo form (393 $Å^2$) [14,18,21]. A comparison of the degree-of-openness values between $MamM_{BW-1}$ (Cα-Arg215, Pro231, and Arg215) (36.7˚) to those in other CDF structures indicates that the dimer showed closer similarity to the closed metal-bound structures of YiiP (~30˚) and previous structures of $MamB_{QH-2}$ (~36˚) and $MamM_{MSR-1}$ (~45˚) [14,15].

In contrast to $MamM_{BW-1}$, $MamB_{BW-1}$ crystal structure exists as a monomer and does not create the classical V-shape dimer in the crystal, yet superimposition of $MamB_{BW-1}$ monomers over $MamM_{BW-1}$ V-shape dimer implies that $MamB_{BW-1}$ may in high probability form a V-shaped dimer under the right conditions (Fig 3B). $MamB_{BW-1}$ dimerization interface, similarly to all CDF-CTD structures, showed the typical parallel S-shaped backbone structure [14,17,18,21,28]. Moreover, it was previously shown that hydrophobic contacts hold the dimeric interactions. Support for such dimerization can be found at the possible interaction between Val235-Pro231 and Ile233-Thr234 from each monomer (Fig 3B). Additional support is given by the $MamM_{BW-1}$ and $MamB_{BW-1}$ electrostatic potential maps (Fig 4), showing that the top of $MamB_{BW-1}$ monomer and the top of $MamM_{BW-1}$ V-shaped dimer have hydrophobic and positive patches, while the lower parts have hydrophobic and negative electrostatic charges (Fig 4C and 4D). Such electrostatic distribution can fit the suggested CTD dimerization when the interaction between the CTD-TMD and the CTD- magnetosome membrane stabilizes and concentrates the proteins and leads to their dimerization. Moreover, similar electrostatic distribution can also be found in Alphaproteobacteria $MamB_{QH-2}$ and $MamM_{MSR-1}$ structures (Fig 4A and 4B). As an alternative, $MamB_{BW-1}$ different oligomeric state can also be explained

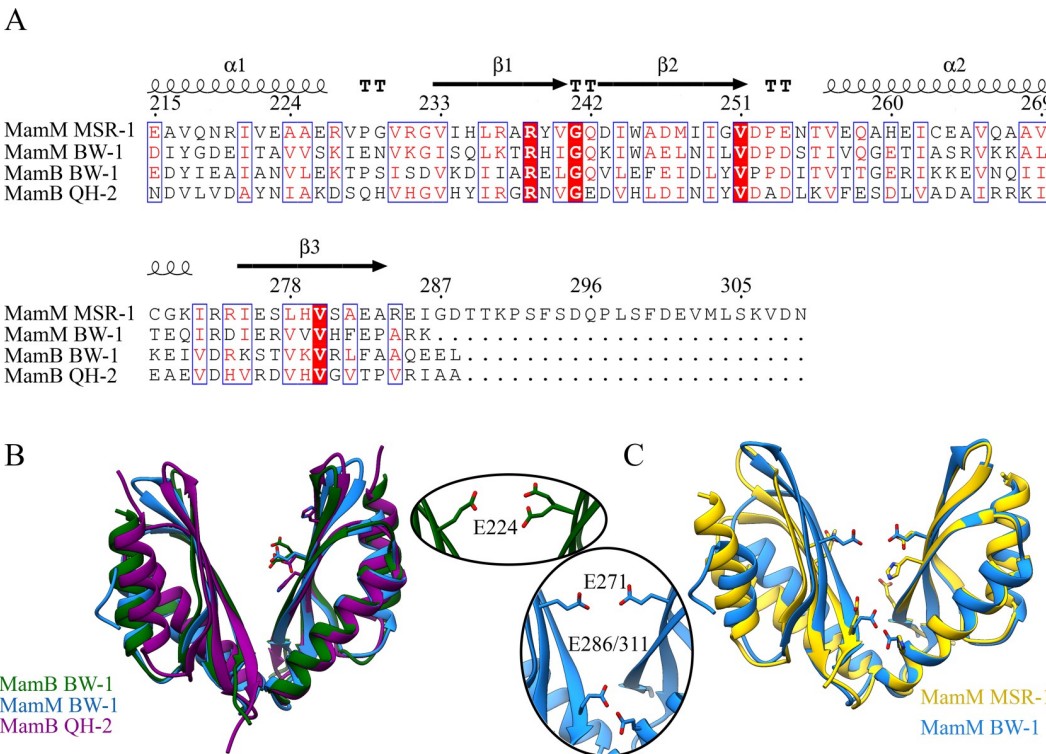

**Fig 1. Cation diffusion facilitators comparison.** (A) Multiple sequence alignment of MamB and MamM CTDs from the *Alphaproteobacteria* (MSR-1 and QH-2) and *Deltaproteobacteria* BW-1. Secondary structure representative, base on MamM$_{MSR-1}$ structure (PDB code 3W5Y). The blue and red frames highlight conserved sequences. (B) Structural overlay of MamB$_{BW-1}$, MamM$_{BW-1}$ and MamB$_{QH-2}$ CTD apo-form structures (PDB codes: 6QFJ, 6QEK, and 5HO5, respectively). Residues that are suggested to participate in the central metal ion-binding site represented as sticks. (C) Structural overlay of MamB$_{BW-1}$ and MamM$_{MSR-1}$ CTD structures (PDB codes: 6QFJ and 3W5Y, respectively) Residues that hypothetically participate in central and peripheral metal ion-binding sites represented as sticks.

by crystal contact and crystallization conditions that break the dimer and create alternative contacts with similar or lower energy.

## Structural conservation of CDF proteins in magnetotactic Proteobacteria

In agreement with previous phylogenies of Proteobacteria, our phylogenetic tree reconstructed from riboproteins confirmed that the magnetotactic strain BW-1, belongs to a class evolutionary divergent and ancestral to that of magnetotactic strains QH-2 and MSR-1 [36] (Fig 5). Using representative MTB strains of the Alphaproteobacteria, Deltaproteobacteria and *Candidatus* Etaproteobacteria, we inferred the MamB and MamM phylogenies individually using FieF sequences as an external outgroup and observed an apparent congruence between species and trees phylogenies for each protein (S2 Fig). This congruence supports that each paralog has emerged from a common MTB ancestor after a very ancient duplication event. This scenario is thus in agreement with previous studies [36]. However, because the internal branches of the trees were very long, we suspected that this scenario and its good statistical support could have been artifactual and have resulted from the long branch attraction of the Deltaproteobacteria towards the external group. We thus build a tree based on the alignment of the three paralogs to possibly reduce this artefact (Fig 6). This analysis led to surprising conclusions; both Maximum-likelihood and Bayesian trees support that MamB emerged from a duplication of MamM in Deltaproteobacteria and was then transferred to other Proteobacteria

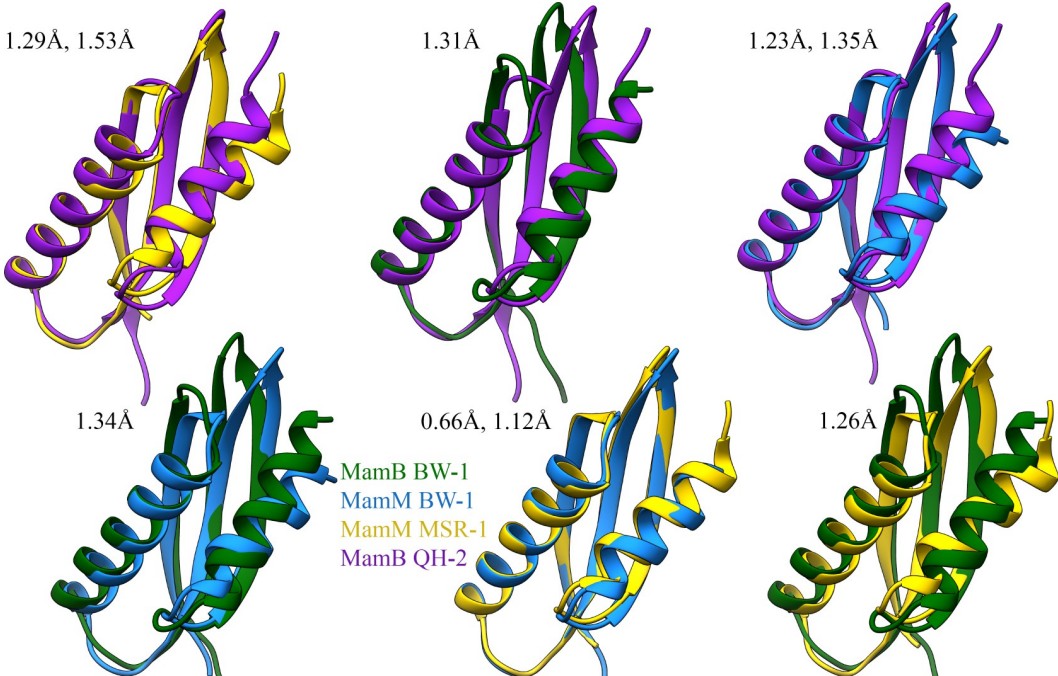

**Fig 2. Comparison of MamB-MamM crystal structures.** Structural overlay of MamB and MamM CTD monomer apo-form structures from *Alphaproteobacteria* and *Deltaproteobacteria* species (PDB codes: 6QFJ (blue), 6QEK (green), 5HO5 (purple) and 3W5Y (yellow)). Root-mean-square deviation (RMSD) values calculated for monomers and dimers respectively, using Swiss-PDB-Viewer [35]. Overall, MamM$_{BW-1}$ and MamM$_{MSR-1}$ share the highest structural similarities with RMSD of 0.66 Å over 77 common backbone atoms.

classes by one or several horizontal gene transfers. According to our analyses, emergence of MamB and its secondary function in vesicle invagination were thus more recent than its paralog MamM. Further analyses with a broader taxonomic sampling will be needed to validate this result.

Despite the high evolutionary divergence of these homologous proteins, we compared the protein structures obtained for BW-1 to those obtained from the analysis of two Alphaproteobacteria species to determine their conservation degree. The structural and functional characterization of MamB$_{QH-2}$ and MamM$_{MSR-1}$ metal-binding sites in previous studies showed that both proteins share a central-binding site (C.B) located at the center of the dimer cavity [14,21,28] C.B within MamB$_{QH-2,}$ composed of His245, Asp247, and His283, and at the same positions within the homolog MamM$_{MSR-1}$ located at Trp247, Asp249 and His285 respectively (Trp247 may be involved due to pi interactions [37]). Besides, MamM$_{MSR-1}$ also has a peripheral metal-binding-site (P.B), composed of Glu289 and His264 from each monomer, and was suggested to be more significant for protein regulation than the C.B [28]. Although the protein sequences show a certain variance, the putative MamB$_{BW-1}$ and MamM$_{BW-1}$ C.B and MamM$_{BW-1}$ P.B are located at the same physical location as described for Alphaproteobacteria, yet minor changes in the amino acid composition can be found. Putative BW-1 C.B is composed of Glu271 (MamM$_{BW-1}$) or Glu224 (MamB$_{BW-1}$) from each monomer. These locations are equivalent to Asp 249 (MamM$_{MSR-1}$) and Asp247 (MamB$_{QH-2}$) (Fig 2B) [14,28]. Trp269, Val307 (MamM$_{BW-1}$) and Glu222, Lys260 (MamB$_{BW-1}$) are also located at the hypothetical metal-binding site but are not predicted to be involved with metal chelation. In MamM$_{BW-1}$ the putative P.B is composed of Glu311 and Glu286 from each monomer (to MamM$_{MSR-1}$ Glu289 and His264 respectively) [28] (Fig 2C). Moreover, these putative metal-binding sites

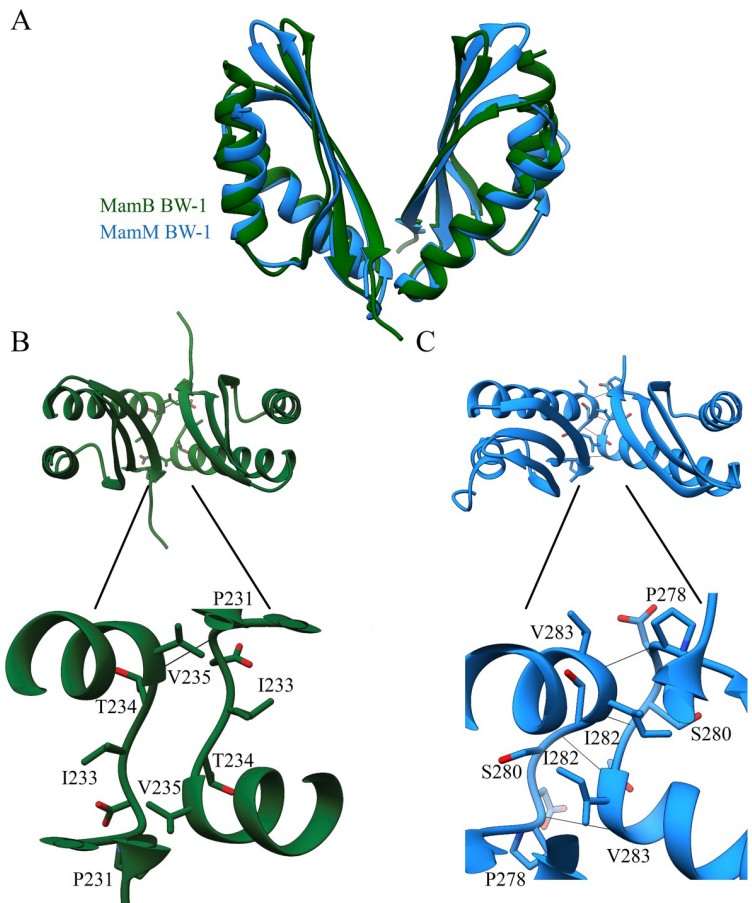

**Fig 3. MamB$_{BW-1}$ and MamM$_{BW-1}$ CTD structures and dimerization interface.** (A) MamB$_{BW-1}$ folded to a metallochaperone-like fold to create a V-shaped dimer. Overlap of the MamM$_{BW-1}$ monomers on MamB$_{BW-1}$ V-dimer structure (PDB codes: 6QFJ (Green), 6QEK (blue). (B) MamB$_{BW-1}$ may create a stable V-shape dimer while the parallel S-shaped typical backbone structure, revealed in the dimerization interface after overlapping the monomers on MamM$_{BW-1}$ dimer. Hydrophobic interactions between Val235-Pro231 and Ile233-Thr234 from each monomer may hold the dimer. (C) MamM$_{BW-1}$ presents a stable dimerization interface located at the bottom of the V-shaped dimer. Dimerization interface stability rests on symmetrical backbone interaction between Pro278-Val283 and Ser280-Ile282.

are supported by the electrostatic potential map that showed negative-charge patches within MamM$_{BW-1}$ center of the V-shaped dimer cavity and in the periphery above the V-bottom (Fig 4C). Such electrostatic-potential distribution can fit the metal-binding site as shown for MamM$_{MSR-1}$ and MamB$_{QH-2}$ (Fig 4A and 4B). Overall, based on our analysis, it can be assumed that the composition of amino acids within MamB$_{BW-1}$ and MamM$_{BW-1}$ putative metal-binding sites are suitable to coordinate and bind metal ions.

To summarize our results, we characterized and determined the crystal structures of MamB and MamM proteins from a Deltaproteobacteria species ancestral to the Alphaproteobacteria species from which homologous proteins were originally characterized. Overall, all the different analyses indicate that MamB$_{BW-1}$ and MamM$_{BW-1}$ share high structural conservation with the well-characterized Alphaproteobacteria proteins. Therefore, it is reasonable to hypothesize that these proteins will have a similar function within the Deltaproteobacteria species. Although unlike previous studies, different protein oligomerization states were observed, and we assume that within the bacteria the high protein concentration, ion presence and other membrane-protein and protein-protein interactions will stabilize MamB and MamM as

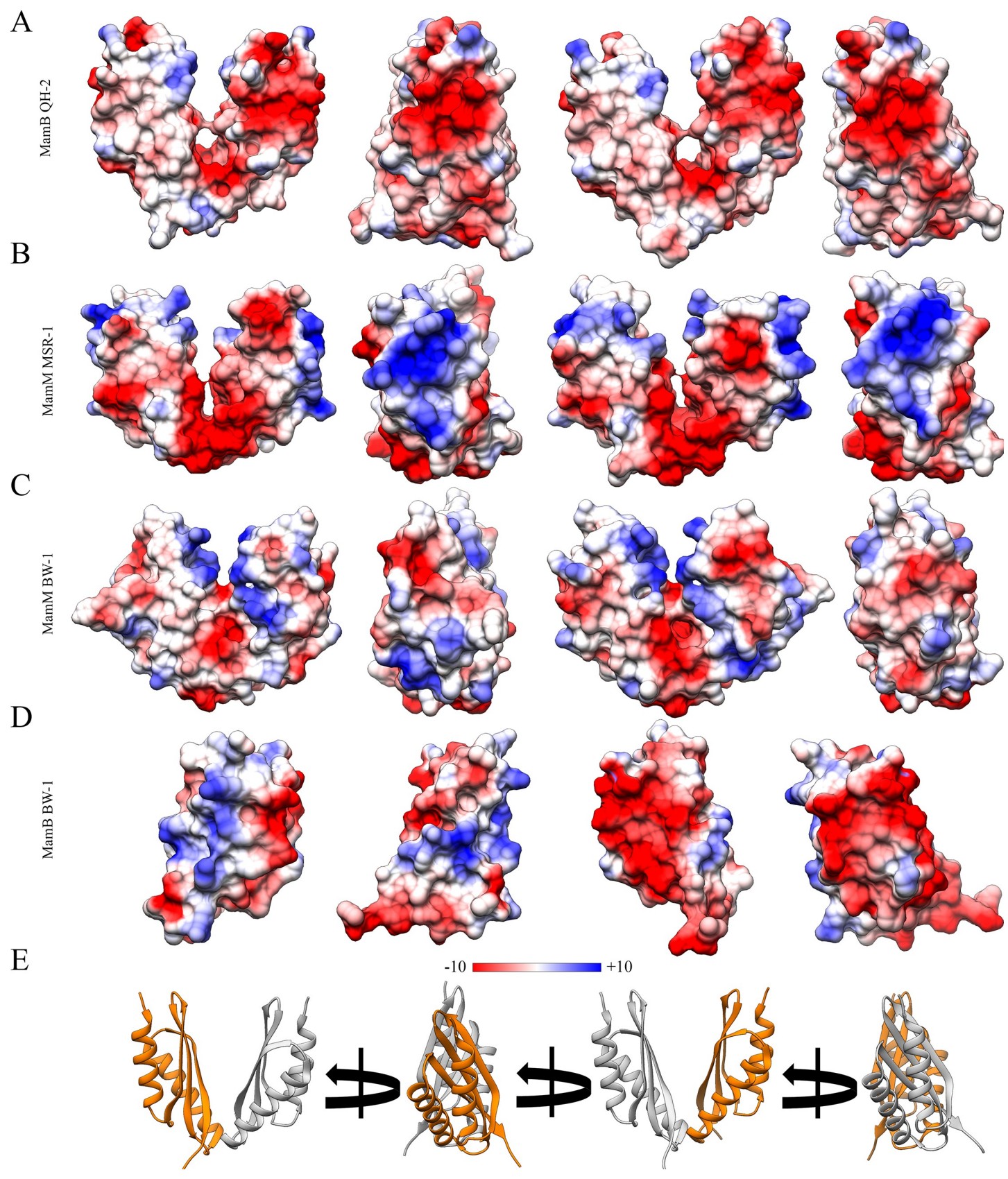

**Fig 4. Proteins CTDs electrostatic potential map.** Alphaproteobacteria and Deltaproteobacteria MamB and MamM CTDs crystal structures, electrostatic potential maps based on the PDB codes: 3HO5, 3W5Y, 6QFJ, and 6QEK. Top of the structures has hydrophobic and positive patches that may fit the interaction model between the CTD-TMD and CTD-magnetosome membrane. Hydrophobic and negative electrostatic charge distribution located at the bottom of the structures. The Central V-shaped dimer cavity showed negative-charge patches found in correlation with the central metal-binding site. MamM structures present an additional negative-charge patch in the periphery above the V-bottom that correlated with the peripheral metal-binding site.

dimers. Further to this work, it is still unknown whether BW-1 MamB and MamM from the greigite cluster will also show structural and functional conservation, and how they relate evolutionarily to their homologs. Such future work will help to understand how a complex organelle like the magnetosome evolved and synthesized.

## Experimental procedures

### Bacterial strains and plasmids for in vitro characterization

Bacterial strains and plasmids used in this study are listed in S4 Table. All strains were cultivated in Luria broth (LB, *E. coli*), as described previously [14].

### Phylogenetic analyses

The genome of 47 complete genomes representing magnetotactic and non-magnetotactic Deltaproteobacteria and Proteobacteria were selected and downloaded from the public database NCBI (https://www.ncbi.nlm.nih.gov). This database was completed with the genome of BW-1 strain and that of 7 species representing the Epsilonbacteraeota [38], a novel phylum proposed as ancestral to Proteobacteria [38,39] composed of the Epsilonproteobacteria and the Desulfurellales. The sequences of 53 bacterial riboproteins [40] (RP) were retrieved from all genomes using the HMMER 3.2.1 software [41,42] and the hidden Markov model (HMM) profiles available in PFAM database [43] (http://pfam.sanger.ac.uk/) to reconstruct the species phylogeny. Paralogous sequences were excluded. RPs were individually aligned with the MAFFT software [44] and trimmed using Gblocks software [45] before their concatenation into a single alignment of 6 442 positions among which 5 771 were polymorphic. All RPs were present in at least 90% of the genomes. A Maximum-Likelihood tree was built with the IQ-TREE software [46] and the model LG+R7 for describing amino-acids evolution that was selected using ModelfFinder [47] and the BIC criterion. 500 replicates of a non-parametric bootstrap approach were conducted to test the robustness of the tree topology.

We also investigated the evolutionary relationships between MamM and MamB of the representative magnetotactic Proteobacteria characterized in this study. MamB and MamM homologs in MTB strains MSR-1, QH-2, IT-1, MC-1, BW-1 and RS-1 (accessions numbers given in S5 Table) [42]) were aligned individually and together with the MAFFT software [42], then trimmed using the BMGE 1.12 software [48] (with -b 3 -g 0.5 options). Three maximum-likelihood trees were built with the IQ-TREE software [46] and a substitution model selected using ModelfFinder [47] and the BIC criterion. Since the gene *fieF* was the genetically closest paralogous gene of the cation diffusion facilitator protein family, three sequences detected in the MTB genomes were used as an external group. For the MamB-MamM tree showing the evolutionary relationships between the two paralogs, a Bayesian approach was also used to build the phylogeny. For that purpose, we used MrBayes version 3.2.7 [49]. This program explores a large set of substitution models through MCMC sampling (e.g. Poisson, BLOSUM62, WAG). In this approach, each model contributes in proportion to the posterior probability distribution of trees. Site heterogeneity was also taken into account by introducing a Gamma correction with four classes. One million iterations were performed and convergence was estimated

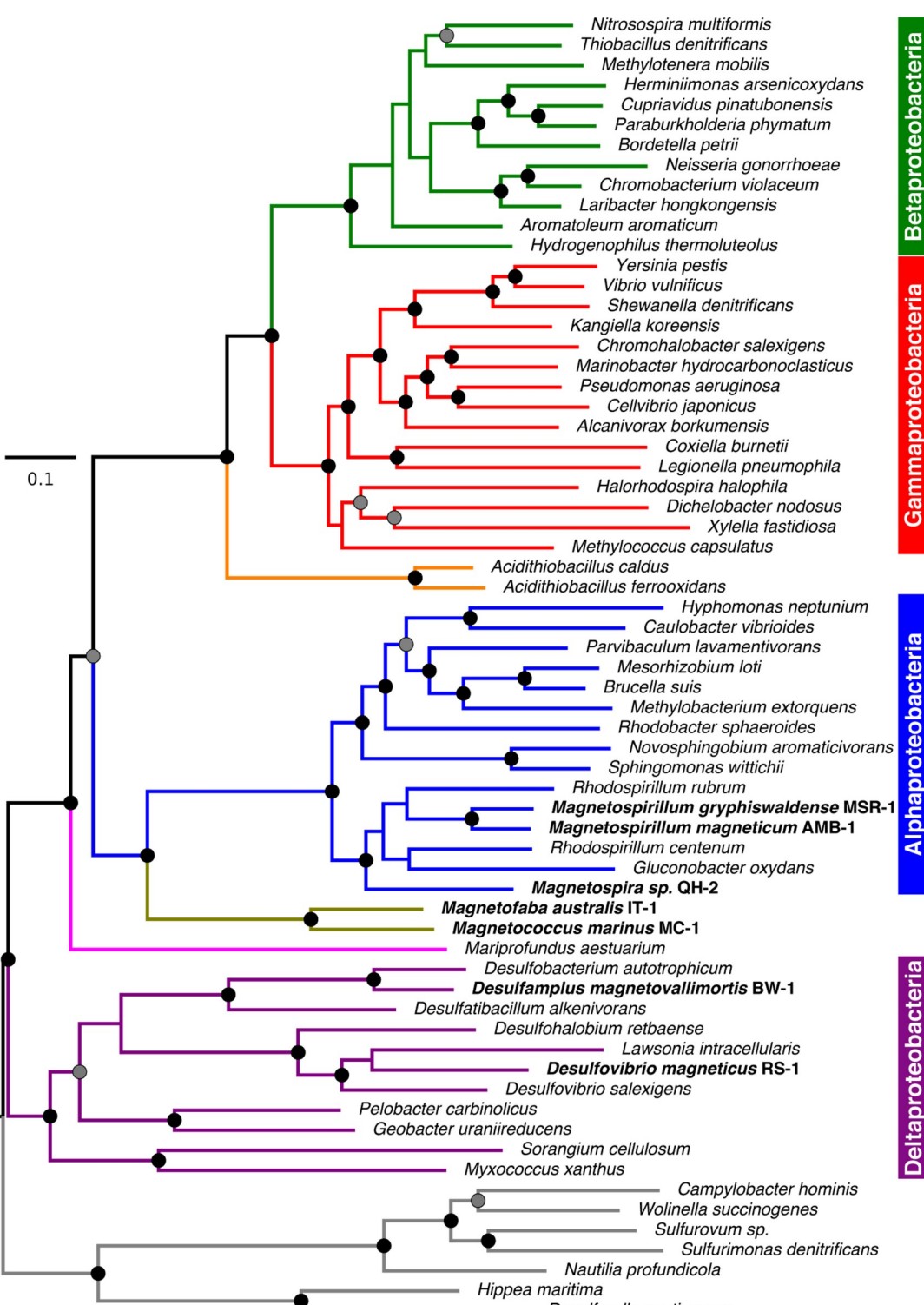

**Fig 5. Phylogenetic tree showing the evolutionary relationships between magnetotactic bacteria and other non-magnetotactic Proteobacteria species.** The tree was built using the Maximum-Likelihood method implemented in IQ-TREE and the concatenation of 53 ribosomal proteins. 500 replicates of a non-parametric bootstrap approach were conducted to test the robustness of the tree topology. Internal branches with support superior to 95% are annotated with a circle. Support values superior to 70% are associated with a grey circle while those below that value are not shown. Magnetotactic species names are in bold. The branch length represents the number of substitutions per site.

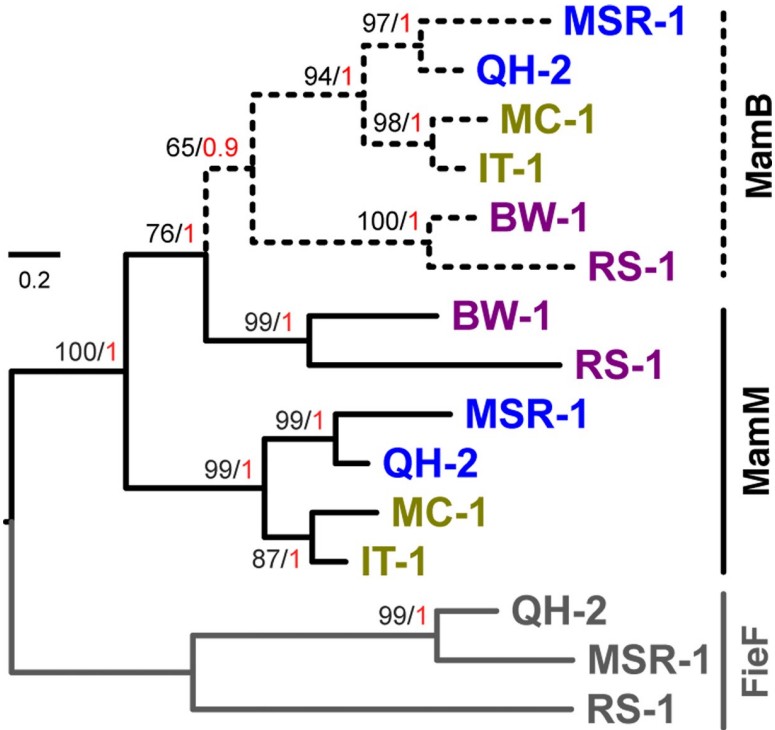

**Fig 6. Maximum-likelihood tree showing the evolutionary relationships between MamM and MamB proteins involved in magnetite biomineralization within magnetotactic Proteobacteria strains.** The tree was built using the Maximum-Likelihood method implemented in IQ-TREE and the trimmed alignment of FieF, MamB and MamM sequences detected in the 6 strains. Color of the names of the strains correspond to their affiliation given in the species tree ([Fig 5]): Alphaproteobacteria (blue), Ca. Etaproteobacteria (brown) and Deltaproteobacteria (Purple). The branch length represents the number of substitutions per site. The robustness of the tree topology was tested with 500 replicates of a non-parametric bootstrap approach (black values). The posterior probability of each clade was also inferred with a Bayesian approach implemented in MrBayes (red values).

using the potential scale reduction factor test. Posterior probabilities at nodes were determined after discarding a burn-in fraction of samples from the chain of 25%.

## Expression of Candidatus Desulfamplus magnetomortis BW-1 MamB and MamM in E. coli

*mamB and mamM* CTD$_{BW-1}$ genes were synthesized and ligated between NdeI and BamHI restriction sites of the pET28a (+) vector (Novagen). In these constructs, the genes were fused with DNA encoding a 6xHis tag at the N-terminus of the proteins, followed by a thrombin proteolysis site (construction by Biomatik, Cambridge, ON, Canada). An *E. coli* Rosetta strain cells harboring plasmid pET28a-MamB-CTD-BW-1 and pET28a-MamM-CTD-BW-1 were grown in auto-induction medium [50] containing kanamycin (100 mg mL$^{-1}$) and chloramphenicol (30 mg mL$^{-1}$) at 37˚C for 8 h. The cultivation temperature was then reduced to 27˚C for a further 48 h. The cells were harvested by centrifugation at 7,438 g for 10 min at 4˚C.

## Purification of Candidatus Desulfamplus magnetomortis BW-1 MamB and MamM

Protein CTD-expressing cells were suspended in buffer A (50 mM Tris–HCl, pH 8, 300 mM NaCl, 20 mM imidazole, 5 mM β-mercaptoethanol, 0.01% Triton X-100) and incubated with

DNase I (10 mg mL$^{-1}$) and protease inhibitor cocktail [100 μM phenylmethylsulfonyl fluoride (PMSF), 1.2 μg mL$^{-1}$ leupeptin and 1 μM pepstatin A] for 20 min at 4˚C. The cells were then disrupted by three cycles in a French press pressure cell at 207 MPa. Cell debris was separated by centrifugation at 43,146 g for 2 h at 4˚C and the soluble fraction was applied onto a home-made gravity-flow Ni–NTA column (4 mL bed volume, 2.5 cm diameter; Bio-Rad Econo-Column chromatography column, Thermo Scientific HisPur Ni–NTA resin) pre-equilibrated with buffer A. The protein was washed with 50 mL buffer B (20 mM Tris–HCl, pH 8, 500 mM NaCl, 40 mM imidazole, 5 mM β-mercaptoethanol). The protein was washed again with buffer C (20 mM Tris–HCl, pH 8, 150 mM NaCl, 40 mM imidazole, 5 mM β-mercaptoethanol), and eluted with buffer D (20 mM Tris–HCl pH 8, 150 mM NaCl 500 mM imidazole and 5 mM β-mercaptoethanol). The protein was concentrated to volume of 350 μL using an Amicon Ultra-cel (3 kDa cutoff, Millipore) at 4˚C and applied onto a size-exclusion column (HiLoad 26/60 Superdex 75, GE Healthcare Biosciences) pre-equilibrated with buffer E (20 mM Tris–HCl pH 8, 150 mM NaCl and 5 mM β-mercaptoethanol). Purified MamM and MamB CTD were then concentrated to 31.4 and 20 mg mL$^{-1}$ respectively, for crystallization, flash-cooled in liquid nitrogen and stored at -80˚C. Throughout the purification process, protein concentrations were determined by spectrometric absorption at 280 nm using a calculated extinction coefficient of 0.493 M$^{-1}$ cm$^{-1}$. Sample purity at this stage was analyzed by sodium dodecyl sulfate-polyacrylamide gel electrophoresis (SDS–PAGE).

## Small Angle X-ray Scattering (SAXS) measurements and analysis

SAXS experiments were also performed at the BM29-BioSAXS beamline at the European Synchrotron Radiation Facility (ESRF) in Grenoble, France. The energy of 12.5 kV corresponding to a wavelength of 0.998 Å$^{-1}$ was selected. The scattering intensity was recorded using a Pilatus 1M detector, in the interval $0.004 < q < 0.5$ Å$^{-1}$, where q is defined as $q = 4\pi/\lambda * \sin\theta$, $2\theta$ is the scattering angle and $\lambda$ is the radiation wavelength. Ten frames with 2 second exposure times were recorded for each sample. Measurements were performed in the flow mode, where samples were pumped through the capillary at a constant flow rate. The dedicated beamline software BsxCuBe and EDNA were used for data collection and initial processing. Further analyses and final plot preparations were performed using IGOR Pro [51] (WaveMetrics, Portland, OR) and the ATSAS suite [52]. The radius of gyration (Rg) was evaluated using the Guinier approximation [53].

## SEC-multiangle light scattering (SEC-MALS)

Purified MamB and MamM protein samples (10 mg/ml) were loaded onto superdex Increase 75 10/300 GL column (GE Healthcare Biosciences), pre-equilibrated with buffer E and connected in line with miniDAWN triple-angle MALS light-scattering detector coupled to an interferometric refractometer (WYATT Technologies, Santa Barbara, CA). Data analysis was done in real-time using ASTRA (Wyatt Technologies, Santa Barbara, CA) and molecular masses were calculated using the Debye fit method.

## Non-specific in-vitro lysine methylation

Pure protein sample of MamM$_{BW-1}$ CTD (5 mg mL$^{-1}$) within buffer E was applied into dialysis tubing for 48h at 4˚C (3.5 kDa cutoff, Thermo scientific) against 4 L buffer F (10 mM HEPES pH 7.5 and 150 mM NaCl). Methylation reaction mix was added at two intervals of 2h incubation at 4˚C (20 μL of 1.018 M Dimethylamine borane complex and 40 μL of 1 M Formaldehyde); afterward 10 μL of 1.018 M Dimethylamine borane complex was added, and the total reaction mix was incubated overnight at 4˚C. Precipitant was separated by centrifugation at

25,000 g for 5 min at 4˚C, 125 μL of 1M Tris pH 7.5 were added to the soluble fraction to stop the reaction. Followed with centrifugation, the methylated protein was applied onto a size-exclusion column (HiLoad 26/60 Superdex 75, GE Healthcare Biosciences) pre-equilibrated with buffer E, at a flow rate of 4 mL min$^{-1}$. Purified methylated- MamM$_{BW-1}$ CTD was then concentrated to 13.73 mg mL$^{-1}$ for crystallization, flash-cooled in liquid nitrogen and stored at -80˚C.

### BW-1 MamB and MamM CTDs crystallization

MamB$_{BW-1}$ and methylated MamM$_{BW-1}$ CTDs were crystallized using the sitting-drop vapor-diffusion method at 20˚C. Accordingly, 0.3 μL protein solution (in buffer E) and 0.3 μL reservoir solutions containing (A) 20% PEG 3,350, 0.1 M Tris pH 8.6, 0.2 M Ammonium acetate (B) 25% PEG 3,350, 0.1 M Bis-Tris pH 5.5 and 0.2M sodium chloride respectively were mixed to form the drop. The crystals diffract up to 2.13 Å and 1.95 Å.

### Diffraction data collection and structure determination

MamB$_{BW-1}$ and methylated MamM$_{BW-1}$ CTDs crystals were harvested and flash-cooled in liquid nitrogen with respectively 0.1 μL 50% PEG 3350 or 1 μL from crystallization condition added to the drop as a cryo-protecting solution. Diffraction data were collected using an image-plate detector system (PILATUS 6M-F 424 x 435 mm$^2$; DECTRIS, Baden, Switzerland and MARmosaic 225 mm CCD; MAR Research, Norderstedt, Germany). Data collection was performed at -173˚C. For the MamB CTD native data set, a total of 206 frames were collected with an oscillation range of 1.75˚ and an exposure time of 10 sec per image. The crystal-to-detector distance was 208.2 mm. For the MamM data set, a total of 2500 frames were collected with an oscillation range of 0.15˚ and an exposure time of 0.037 min per image. The crystal-to-detector distance was 339.56 mm. The data were processed using HKL-2000 [54], XDS [55], Xia2, Aimless and iMosflm from the CCP4i program suite [56]. Molecular replacement was performed using Molrep and PHASER [57,58], against MamB and MamM structures (PDB codes 5HO3 and 5W5X). Structure refinement was accomplished using the programs REFMAC5 [59], PDB_REDO server [60] and Coot [61].

### Structure analysis

Root-mean-square deviation (RMSD) calculations were performed with SwissPDB viewer [35] using the domain alternate fit feature to align structures based on the conserved domain and to define the conformational changes of the structural homologs. Electrostatic potential maps were calculated using APBS and coulombic surface coloring at UCSF Chimera [62]. The dimerization interface was calculated using PDBe PISA server [63]. All structure figures were prepared using UCSF Chimera [62].

#### Structure coordinates

Structures of MamB and MamM (6QFJ and 6QEK respectively) have been submitted to the Protein Data Bank.

## Supporting information

**S1 Fig. BW-1 MamB and MamM size-exclusion chromatogram multiangle light scattering.** SEC-MALLS chromatogram shows the elution curve of BW-1 CTD proteins, refractive index correlated with UV absorbance at 280nm. Molar mass was calculated.
(TIF)

**S2 Fig. Maximum-likelihood trees showing the evolutionary relationships between magnetotactic Proteobacteria strains based on MamM or MamB proteins involved in magnetite or greigite biomineralization.** The trees were built using the Maximum-Likelihood method implemented in IQ-TREE and the trimmed alignment of FieF with MamB or MamM sequences detected in the 6 strains. Color of the strain names correspond to their affiliation given in the species tree (Fig 5): Alphaproteobacteria (blue), Ca. Etaproteobacteria (brown) and Deltaproteobacteria (Purple). The branch length represents the number of substitutions per site. The robustness of the tree topology was tested with 500 replicates of a non-parametric bootstrap approach.
(TIF)

**S1 Table. Small angel x-ray scattering data analysis.**
(DOCX)

**S2 Table. Crystallization of BW-1 MamB and MamM CTDs.**
(DOCX)

**S3 Table. Data collection and refinement statistics.**
(DOCX)

**S4 Table. Bacterial strains, plasmids, and genomes.**
(DOCX)

**S5 Table. Sequences of MamB, MamM and FieF used in this study.**
(XLSX)

## Acknowledgments

We thank ESRF (Grenoble, France) for providing synchrotron radiation facilities beamline ID23-2 and ID14-4 and for assistance during data collection, and Dr. Petra Pernot for assistance in using beamline BM29-bioSAXS. We thank the 1st CCP4/BGU Structural Solution Workshop and especially to Dr. Kay Diederichs and Dr. Michael Yosephov who assisted in data processing and structure determination of MamM-CTD. Without their professional help, we would not have been able to determine this structure. We thank Dr. Anat Shahar from the Macromolecular Crystallography Research Center (Ben-Gurion University of the Negev, Israel). We thank Dr. Hila Nudelman and Dr. Shiran Barber-Zucker for a fruitful discussion and good scientific advice. The authors are grateful to the INRAE Migale bioinformatics platform (https://migale.inra.fr) for providing computational resources.

## Author Contributions

**Conceptualization:** Noa Keren-Khadmy, Natalie Zeytuni, Raz Zarivach.

**Data curation:** Noa Keren-Khadmy, Natalie Zeytuni, Nitzan Kutnowski, Guy Perriere, Caroline Monteil.

**Formal analysis:** Noa Keren-Khadmy, Nitzan Kutnowski, Guy Perriere, Caroline Monteil, Raz Zarivach.

**Funding acquisition:** Raz Zarivach.

**Investigation:** Noa Keren-Khadmy.

**Methodology:** Noa Keren-Khadmy, Guy Perriere, Caroline Monteil, Raz Zarivach.

**Project administration:** Raz Zarivach.

**Supervision:** Guy Perriere, Raz Zarivach.

**Validation:** Raz Zarivach.

**Writing – original draft:** Noa Keren-Khadmy, Caroline Monteil, Raz Zarivach.

**Writing – review & editing:** Noa Keren-Khadmy, Natalie Zeytuni, Nitzan Kutnowski, Guy Perriere, Caroline Monteil, Raz Zarivach.

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
