## [Decision Letter · Decision Letter 0]

5 Feb 2020

PONE-D-19-36027

From conservation to structure, studies of magnetosome-associated cation diffusion facilitators (CDF) proteins in Proteobacteria.

PLOS ONE

Dear Prof. Zarivach,

Thank you for submitting your manuscript to PLOS ONE. It has been sent to three expert reviewers. As you will see in their comments pasted below, the reviewers recommend publication pending minor modifications. Based on their opinion and recommendation, I encourage you to carefully address their comments and invite you to submit a revised version of the manuscript by Mar 21 2020 11:59PM. To enhance the reproducibility of your results, we recommend that if applicable you deposit your laboratory protocols in protocols.io, where a protocol can be assigned its own identifier (DOI) such that it can be cited independently in the future. For instructions see: http://journals.plos.org/plosone/s/submission-guidelines#loc-laboratory-protocols

We look forward to receiving your revised manuscript.

Kind regards,

Eric Cascales

Academic Editor

PLOS ONE

Reviewers' comments:

Reviewer's Responses to Questions

**Comments to the Author**

1. Is the manuscript technically sound, and do the data support the conclusions?

Reviewer #1: Yes

Reviewer #2: Yes

Reviewer #3: Yes

2. Has the statistical analysis been performed appropriately and rigorously? 

Reviewer #1: Yes

Reviewer #2: N/A

Reviewer #3: N/A

3. Have the authors made all data underlying the findings in their manuscript fully available?

Reviewer #1: Yes

Reviewer #2: Yes

Reviewer #3: Yes

4. Is the manuscript presented in an intelligible fashion and written in standard English?

Reviewer #1: Yes

Reviewer #2: Yes

Reviewer #3: Yes

5. Review Comments to the Author

Reviewer #1: This manuscript by Keren-Khadmy et al for the first time characterizes the cation diffusion facilitator MamB and MamM proteins and CTD crystal structures from a proteobacterial magnetotactic bacterium. While the overall structure of these proteins is similar to those of previous CDF-CTD structures their oligomeric states differ. In addition to regular dimeric forms also monomers were observed. Interestingly, also metal binding site aa compositions differ from previous reports.

The experiments are well performed and presented and deserve publication in Plos ONE. However, a few shortcomings (listed below) should be addressed before publication.

L59: „several chains“ – not true for all MTB

L61: oxic-anoxic transition zone better than microoxic transition zone

LL61: “Most of the genes crucial for the magnetosome biogenesis are specific to MTB” - might be misleading since homologs to mam genes exist outside of MTB as well.

L63: the bacterial genome of their genomes

LL65-67: MamB and MamM are conserved among all MTB (including more phyla than listed here).

LL67: These sentences imply that BW-1 has MamM and B for greigite formation only. I would recommend to modify the whole paragraph to clarify the phylogenetic occurrence of MamB and MamM and the presence of two different magnetosome gene clusters in BW-1 (of which each contains mamB/mamM homologs).

L72: use bacterial instead of bacterium

L75: insert a before “cytosolic”

L79: grammar

LL96: what are the consequences of this alternative dimer packing? Are proteins still functional?

LL101: avoid “additionally” and “additional“ within the same sentence.

L104: delete class

LL145: “Third, we assume that in-vivo MamB and MamM local-concentration might be higher as the magnetosome contains only a few protein complexes that involve MamB and MamM.” – How can the MamB and MamM concentration increase at the magnetosome when it contains only a few complexes?

LL151: Previously it was reported that MamM and MamB directly interact with each other. It would therefore also be possible that heterodimeric MamM/B complex are more preferred. This should be at least discussed here.

LL193: Why does the apo MamM-BW-1 dimer resemble other metal-bound CDF-CTDs?

LL237: additional trees based on MamM and/or MamB protein sequences would be more informative. Especially since the authors stated in their introduction that their results also indicate that these proteins were under high selection pressure over MTB evolution and likely coevolved in Proteobacteria.

LL261: what intrigued me the most is the apparent “flexibility” of the aa composition of the metal binding sites within the different CDF-CTDs. I think it would be a great benefit for the manuscript if the authors could add paragraph to discuss and analyze metal-binding site variations in a broader way (e.g. is there conservation among certain groups?).

Fig.1: Numbering of alignment in A and highlighted metal binding site in B do not match (Position 224 of MamB BW-1 in A is a Val not a Glu as depicted in B). Why are there two residues shown on the right monomer in the inset of fig 1B but only one aa label?

Bibliography: Ref 19 + 25 and 12 + 30 refer to the same articles

Reviewer #2: Proteins of the CDF (cation diffusion faciliator) protein family, which are in fact proton-cation-antiporters, transport transition metal cations across biological membranes. The first CDF protein sequences came from yeast (Kamizomo) and a beta-proteobacterium (Nies), the protein family was defined by Saier and the first structure came from E. coli YiiP (Fu). Genes from CDF proteins were found in a gene region responsible for the formation of magnetosomes (Schueler) at a time when first evidence was presented that YiiP may be an iron efflux system re-named as FieF (Grass). Uebe et al described the role of two magnetosome-associated CDFs in magnetite biomineralization.

The authors produced and crystallized the carboxy-terminal domains of the two magnetosome-associated CDFs MamM and MamB from a deltaproteobacterium. These carboxy-terminal domains are compared with each other and the respective domains from two orthologs from alphaproteobacterial. Possible amino acyl residues involved in metal binding and dimerization are discussed. This allows the first direct comparison of a MamM with a MamB from the same organism.

1. Line 47. Many abbreviations are used only a few times and should not be used. Abbreviations of bacterial species names should not be listed as abbreviation since this is standard operation procedure. If listed, they should be in italics. If LB is defined, Luria with capital “L”.

2. Line 64. Cite the source for the name CDF or TC category.

3. Line 108. When mentioned a second time, the genus name of a bacterial species name must be abbreviated.

4. Line 123. Only genes are expressed, proteins are produced.

5. Line 127, too many “we” for a scientific paper.

6. Line 134, to demonstrate that I have read it: two commas.

7. Line 139. The hypothesis that the transmembrane part of the CDF contributes to the formation of the dimers comes at line 145. Nevertheless, differentiate here between the references citing dimeric full-length proteins and dimeric isolated domains.

8. Line 158 and methods. I know some protein-structure people who have foam around their mouth if a structure was obtained from a protein isolated using a His-tag, because this tag may influence the structure heavily. I would not go that far. Nevertheless, since the His-tag was followed by a thrombin recognition sequence, why was the His-tag not removed after the protein purification? The risk of an artifact would be decreased.

9. It seems to be the PLOS format but having the legend within the text is strange and interrupts the reading flow.

10. Line 190. What are the forces used for the interaction of hydrophobic amino acyl residues such as Val and Ile with the helix breaker Pro and the polar Ser?

11. Figure 5. I do not see why the deltas should be ancestral to the alphas.

12. Line 274. This seems to be the conclusion. If the main message of this paper is the direct comparison of the structures of the two different magnetosome-associated CDFs, what did we learn? It all looks the same.

13. Line 362. What is the sense of this methods chapter?

14. Fig. 1. There is no E224 in MamB BW-1 but a Val. Why is the shown residue duplicated in one protomer? Two solutions of the same amino acyl residue?

15. Figure 2. The message of this figure is: no interesting differences.

16. Figure 3. Again, how are these amino acid residues supposed to interact? Van der Waals? Hydrogen bonds? Not from a Val residue.

Reviewer #3: In this paper, Keren-Khadmy et al, determined crystal structures of C-terminal domains (CTDs) of MamB and MamM from Desulfamplus magnetovallimortis strain BW-1. Overall, this paper is well written contains interesting results.

The authors discussed about structural conservations of CTDs of cation diffusion facilitator proteins in magnetotactic bacteria. The works on structural conservation in magnetosome-associated proteins is still very limited. I think that this paper has value to publish for accumulating molecular information about magnetosome synthesis.

6. PLOS authors have the option to publish the peer review history of their article (what does this mean?). If published, this will include your full peer review and any attached files.

Reviewer #1: No

Reviewer #2: No

Reviewer #3: No

---

## [Author Response · Author response to Decision Letter 0]

31 Mar 2020

Response to reviewers is attached as a detailed file.

---

## [Editor Report · Decision Letter 1]

2 Apr 2020

From conservation to structure, studies of magnetosome-associated cation diffusion facilitators (CDF) proteins in Proteobacteria.

PONE-D-19-36027R1

Dear Raz,

Thank you for submitting your revised version, and for addressing all the minor points raised by the referees. I am pleased to inform you that your manuscript has been judged scientifically suitable for publication. Please note that it will be formally accepted for publication once it complies with all outstanding technical requirements.

With kind regards,

Eric

Eric Cascales

Section Editor

PLOS ONE
---

## [Editor Report · Acceptance letter]

6 Apr 2020

PONE-D-19-36027R1 

From conservation to structure, studies of magnetosome associated cation diffusion facilitators (CDF) proteins in Proteobacteria

Dear Dr. Zarivach:

I am pleased to inform you that your manuscript has been deemed suitable for publication in PLOS ONE. Congratulations! Your manuscript is now with our production department. 

With kind regards,

on behalf of

Dr. Eric Cascales 

Section Editor

PLOS ONE